# DNA-RNA hybrids at DSBs interfere with repair by homologous recombination

Pedro Ortega, José Antonio Mérida-Cerro, Ana G Rondón,
Belén Gómez-González*, Andrés Aguilera*

Centro Andaluz de Biología Molecular y Medicina Regenerativa (CABIMER),
Universidad de Sevilla-CSIC-Universidad Pablo de Olavide, Seville, Spain

**Abstract** DNA double-strand breaks (DSBs) are the most harmful DNA lesions and their repair is crucial for cell viability and genome integrity. The readout of DSB repair may depend on whether DSBs occur at transcribed versus non-transcribed regions. Some studies have postulated that DNA-RNA hybrids form at DSBs to promote recombinational repair, but others have challenged this notion. To directly assess whether hybrids formed at DSBs promote or interfere with the recombinational repair, we have used plasmid and chromosomal-based systems for the analysis of DSB-induced recombination in *Saccharomyces cerevisiae*. We show that, as expected, DNA-RNA hybrid formation is stimulated at DSBs. In addition, mutations that promote DNA-RNA hybrid accumulation, such as *hpr1Δ* and *rnh1Δ rnh201Δ*, cause high levels of plasmid loss when DNA breaks are induced at sites that are transcribed. Importantly, we show that high levels or unresolved DNA-RNA hybrids at the breaks interfere with their repair by homologous recombination. This interference is observed for both plasmid and chromosomal recombination and is independent of whether the DSB is generated by endonucleolytic cleavage or by DNA replication. These data support a model in which DNA-RNA hybrids form fortuitously at DNA breaks during transcription and need to be removed to allow recombinational repair, rather than playing a positive role.

*For correspondence:
gomezb@us.es (BG);
aguilo@us.es (AA)

## Introduction

DNA double-strand breaks (DSBs) are extremely cytotoxic DNA lesions that can arise as a consequence of direct DNA breakage or by replication fork blockage at DNA lesions, including single-stranded DNA (ssDNA) breaks. Two main mechanisms have evolved to repair DSBs: non-homologous end joining (NHEJ), which consists of the direct ligation of the DNA ends after some minimal processing; and homologous recombination (HR), which relies on the homology of a template DNA to bypass the break and complete replication, thus restoring the genetic information at the break. DSBs are preferentially repaired by sister chromatid recombination (SCR), an HR reaction that uses the intact sister chromatid as a template, being key during the S and G2 phases of the cell cycle (*González-Barrera et al., 2003*; *Johnson and Jasin, 2000*; *Kadyk and Hartwell, 1992*). Interestingly, the transcriptional context in which DSBs occur influences the repair pathway choice (*Marnef et al., 2017*). Thus, DSBs occurring within actively transcribed chromatin are preferentially repaired by HR compared to non-transcribed sequences located in euchromatin (*Aymard et al., 2014*; *Wei et al., 2015*).

DNA breaks facilitate the formation of DNA-RNA hybrids in yeast and mammalian cells. Although DNA-RNA hybrids can function in a number of physiological processes, their unscheduled formation is a well-known source of genome instability (*García-Muse and Aguilera, 2019*). In accordance, cells have evolved multiple mechanisms to counteract their formation including RNA coating by RNA biogenesis factors, such as the THO complex, DNA-RNA unwinding by different helicases including Senataxin and UAP56/DDX39B, RNA degradation by RNase H enzymes, as well as DNA repair-

mediated removal (*García-Muse and Aguilera, 2019*). Despite evidence indicating that DSBs promote DNA-RNA hybridization, the effect of such hybrids at DSBs is yet unclear (*Aguilera and Gómez-González, 2017*; *Marnef and Legube, 2021*). Thus, whereas different helicases have been shown to remove DNA-RNA hybrids at DSBs (*Cohen et al., 2018*; *Li et al., 2016*; *Sessa et al., 2021*; *Yu et al., 2020*), other reports postulated that RNA molecules could act as a functional intermediate for DSB repair (*D'Alessandro et al., 2018*; *Keskin et al., 2014*; *Liu et al., 2021*; *Lu et al., 2018*; *Ohle et al., 2016*). In addition, RNase H enzymes have been claimed to either play a crucial role at DSBs (*Ohle et al., 2016*) or no role at all (*Zhao et al., 2018*).

Trying to add light to this question, we have investigated how DNA-RNA hybrids formed at DSBs influence their repair. We used the HO endonuclease (*Kostriken et al., 1983*) or a mutated FLP nickase (Flp-H305L, FLPm) (*Tsalik and Gartenberg, 1998*) to induce either replication-independent or replication-born DSBs, respectively. We observed that plasmids were lost at high frequency upon DSB induction in transcribed DNA regions in *Saccharomyces cerevisiae* mutants with high levels of DNA-RNA hybrids. Hybrids are accumulated co-transcriptionally at sites where breaks are induced and interfere with DSB repair by HR. The data support a model in which unscheduled DNA-RNA hybrids form co-transcriptionally at DNA breaks and need to be removed in order to allow efficient DSB repair and genome stability.

## Results

### High loss of cleaved plasmids in DNA-RNA hybrid-accumulating mutants

To analyze the potential impact that DNA-RNA hybrids putatively accumulated at DSB sites could have on DNA repair, we assessed the frequency of plasmid loss upon break induction in yeast DNA-RNA hybrid accumulating mutants (*Figure 1*). We constructed a set of centromeric plasmids containing the *LEU2* gene under the control of the *TET* promoter (*tetp*), that is strongly repressed upon doxycycline (dox) addition (*Figure 1—figure supplement 1a*), interrupted by either an insertion of the flippase recognition target (*FRT*) or the *HO* site-specific endonuclease sites (*Figure 1a*). Moreover, the *FRT* site was inserted in the two orientations so that the ssDNA break could be specifically induced in either the transcribed (T) or non-transcribed (NT) strand. Galactose-induced overexpression of FLPm or HO induced replication-born DSBs or replication-independent DSBs, respectively, as previously shown (*Kostriken et al., 1983*; *Mehta and Haber, 2014*; *Nielsen et al., 2009*; *Ortega et al., 2019*).

We transformed wild type, *rnh1Δ rnh201Δ* and the *hpr1Δ* mutant of the THO complex, known to co-transcriptionally accumulate high levels of DNA-RNA hybrids (*Cerritelli and Crouch, 2009*; *Luna et al., 2019*), with each of the constructs and quantified plasmid loss after cleavage induction (+Gal) by FLPm or HO and under low or high transcription of the plasmid-born *leu2* allele. For the two FRT constructs, the frequency of plasmid loss under low or repressed transcription of the *leu2* allele (+dox) was below 5% in all strains tested, despite FLPm being expressed for 24 hr (*Figure 1b*). In contrast, under high *leu2* transcription, the frequency augmented to more than 20% in *rnh1Δ rnh201Δ* and *hpr1Δ* cells. The levels of spontaneous plasmid loss under no cleavage induction were barely detectable in any of the strains (*Figure 1—figure supplement 1b*). Thus, *rnh1Δ rnh201Δ* and *hpr1Δ* cells specifically induce the loss of transcribed cleaved plasmids.

For the HO construct, experiments were performed after 1 hr of HO induction, for which the frequency of plasmid loss was already around 30% under low or repressed transcription in all strains (*Figure 1b*). Upon transcription activation, however, wild-type cells reduced the frequency of plasmid loss to 8% (*Figure 1b*). This is in agreement with our previous observation that HO endonuclease activity at the HO site is less efficient at highly transcribed chromatin (*González-Barrera et al., 2002*). Importantly, plasmid loss levels were above 25% in *rnh1Δ rnh201Δ* and *hpr1Δ* mutants (*Figure 1b*). Similar results were obtained after 3 hr of cleavage induction, with levels differing from 25% in wt cells to 55% in the mutants (*Figure 1—figure supplement 1c*).

Therefore, whereas *rnh1Δ rnh201Δ* and *hpr1Δ* cells led to similar loss levels of cleaved plasmids by either FLPm or HO under low transcription of the plasmid-born *leu2* allele, loss frequencies were specifically augmented when *leu2* transcription was induced. Moreover, despite the fact that RNase H1 overexpression can be toxic and cause DNA damage and replicative stress (*Paulsen et al., 2009*;

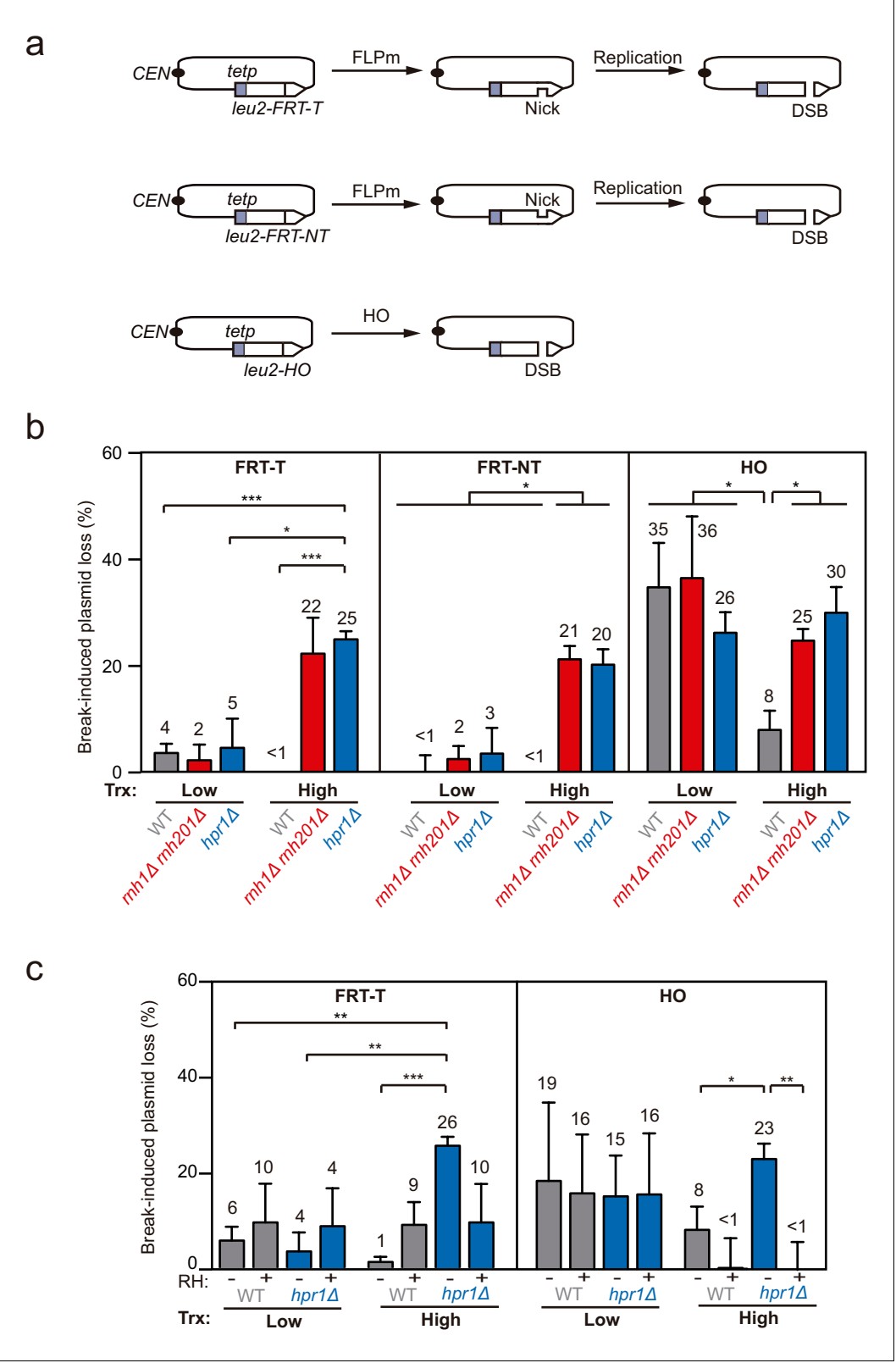

**Figure 1.** Loss of cleaved plasmids in DNA-RNA hybrid-accumulating mutants. (**a**) Scheme of the pCM189-L2FRT-T (upper panel) or -NT (middle panel) and pCM189-L2HO (lower panel) plasmids. FLPm or HO endonuclease induction leads to either nicks that will be converted into DSBs by replication or to replication-independent DSBs, respectively. (**b**) Percentage of break-induced plasmid loss in wild type (WFLP and WS), *rnh1Δ rnh201Δ* (WFR1R2

*Figure 1 continued on next page*

*Figure 1 continued*

and WSR1R2), and *hpr1Δ* (WFHPR1 and WSHPR1) strains transformed with pCM189-L2FRT-T (FRT-T), pCM189-L2FRT-NT (FRT-NT), or pCM189-L2FHO (HO) under low or high transcription and after either 24 hr of FLPm induction or 1 hr of HO induction (n≥3). (c) Percentage of break-induced plasmid loss in wild type (WFLP and WS) and *hpr1Δ* (WFHPR1 and WSHPR1) strains transformed with pCM189-L2FRT-T (FRT-T) or pCM189-L2FHO (HO) and either pRS314 (RH−) or pRS314-GALRNH1 (RH+) under low or high transcription and after either 24 hr of FLPm induction or 1 hr of HO induction (n≥4). Mean and SEM of independent experiments consisting in the median value of six independent colonies each are plotted in (b, c) panels. *p≤0.05; **p≤0.01; ***p≤0.001 (unpaired Student's t-test). See also *Figure 1—figure supplement 1*. Data underlying this figure are provided as *Figure 1—source data 1*. Trx, transcription.

The online version of this article includes the following source data and figure supplement(s) for figure 1:

**Source data 1.** Loss of cleaved plasmids in DNA-RNA hybrid-accumulating mutants.

**Figure supplement 1.** *leu2* expression levels in the TINV-FRT system, spontaneous plasmid loss and break-induced plasmid loss after 3h.

**Figure supplement 1—source data 1.** *leu2* expression levels in the TINV-FRT system, spontaneous plasmid loss and break-induced plasmid loss after 3h.

*Domínguez-Sánchez et al., 2011*), which could contribute to further plasmid loss, we observed that it fully suppressed the increase of plasmid loss induced by *hpr1Δ* in both FRT and HO constructs (*Figure 1c*), indicating that the high loss of cleaved plasmids was caused by DNA-RNA hybrids. These results suggest that DNA-RNA hybrids could interfere with the repair of DSBs, regardless of whether directly generated by an endonuclease or during replication.

## Impaired repair of replication-born DSBs in hybrid-accumulating mutants

To directly study DSB repair upon DNA-RNA hybrid accumulation, we took advantage of the TINV recombination system. This system is based on a centromeric plasmid with two *leu2* inverted repeats, one of which containing an endonuclease site (*González-Barrera et al., 2003*). We first focused on the repair of replication-born DSBs (FLPm-induced), taking advantage of the existence of the previously validated TINV-FRT plasmid (*Ortega et al., 2019*), which we investigated in two versions (FRT-T and FRT-NT) (*Figure 2a*). We measured the appearance of breaks (2.4 and 1.4-kb fragments) by Southern blot after FLPm induction and observed by alkaline gel electrophoresis that ssDNA cleavage reached up to 20% (*Figure 2—figure supplement 1a–b*), but the percentage of DSB molecules was always below 1% as revealed by neutral gel electrophoresis (*Figure 2—figure supplement 1c–d*). The levels of DSBs detected were higher in *rnh1Δ rnh201Δ* and *hpr1Δ* mutants (*Figure 2—figure supplement 1c–d*). This could be a consequence of either a major efficiency of breakage or a lower efficiency of DSB repair, which in these FRT-based constructs occurs preferentially by SCR (*Ortega et al., 2019*). To determine the frequency of SCR, we quantified the events involving an exchange between unequal repeats in the two sister chromatids (unequal sister chromatid exchange, uSCE), which leads to a dicentric dimer intermediate that can be visualized by the 4.7 and 2.9-kb bands resulting from XhoI and SpeI digestion (*Figure 2a*; *González-Barrera et al., 2003*). Given the proximity of the 4.7-kb band to the strong 3.8-kb band arising from the digestion of the more abundant intact plasmid, we relied on the 2.9-kb band to quantify SCR as previously described (*Ortega et al., 2019*). Other recombination reactions are also possible but known to occur at a minor and irrelevant frequency (*Cortés-Ledesma et al., 2007*). Thus, to estimate repair at each time point, we calculated the ratio between the SCR-derived molecules (2.9-kb fragment, *Figure 2a*) and the sum of repaired and cleaved molecules (2.9, 2.4, and 1.4-kb fragments, *Figure 2a*) (see Materials and methods). Under high transcription of the FRT site, a subtle but not significant defect in repair was detected in *rnh1Δ rnh201Δ* cells in both FRT-T and FRT-NT constructs and in *hpr1Δ* cells in the FRT-T construct (*Figure 2b*). Interestingly, such a tendency was not observed under low transcription (*Figure 2c*) suggesting that although subtle, there could be a repair defect that was transcription-dependent.

Even though the SCR intermediate detected physically has to be resolved by an additional HR event to give rise to a recombinant plasmid (*González-Barrera et al., 2003*), the frequency of Leu⁺ recombinants (*Figure 2—figure supplement 2a*) can be used to genetically infer unequal SCR in our

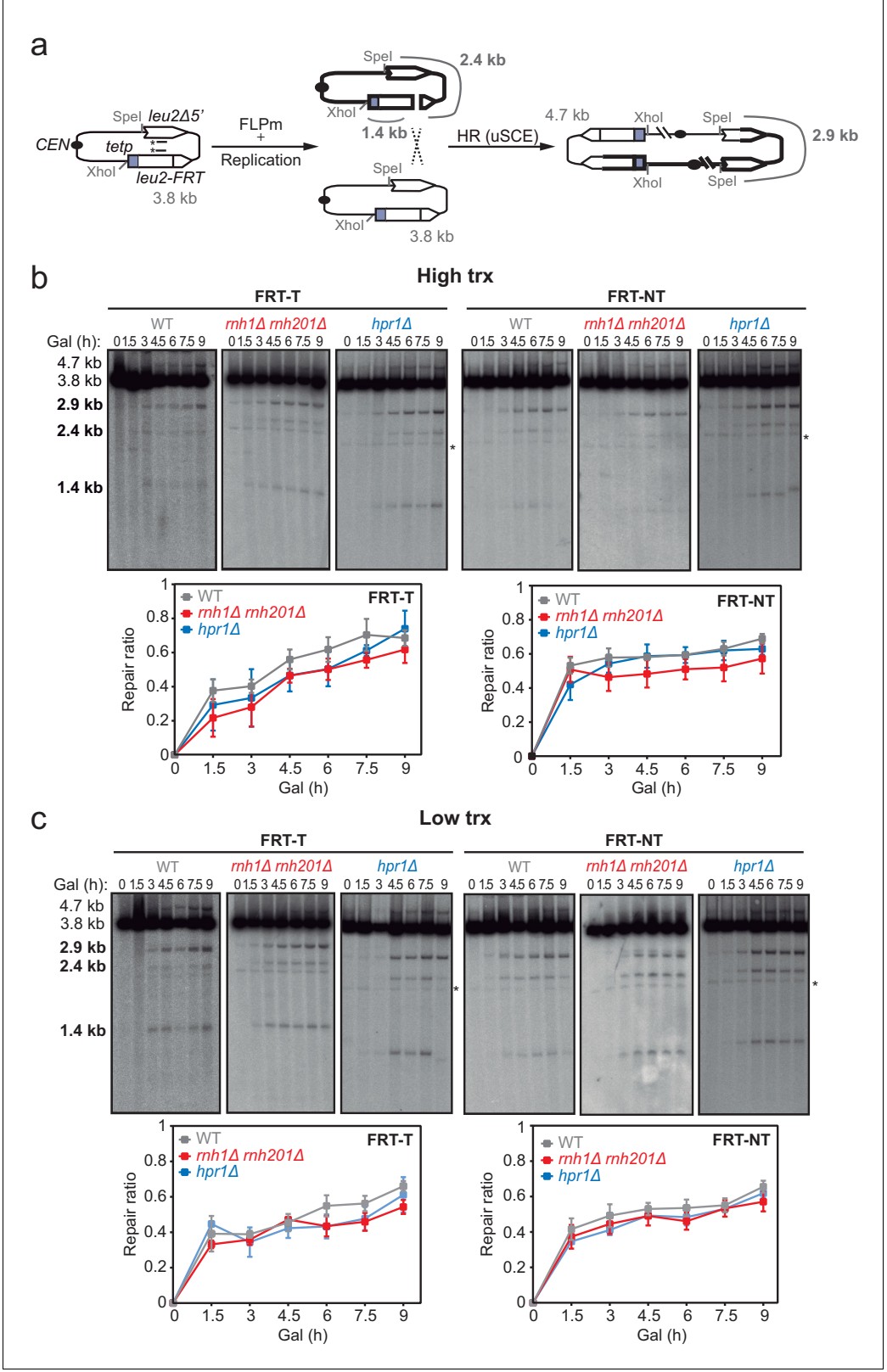

**Figure 2.** Effect of *rnh1Δ rnh201Δ* and *hpr1Δ* in the repair of replication-born DSBs with the sister chromatid. (a) Schemes of the TINV-FRT system, in which FLPm induction of nicks leads to replication-born DSBs in one of the sister chromatids so that the intact sister chromatid can be used as a template for repair. The repair intermediate resulting from sister chromatid recombination (SCR) involving an exchange between unequal repeats in the two

*Figure 2 continued on next page*

*Figure 2 continued*

sister chromatids (unequal sister chromatid exchange, uSCE) is depicted. Fragments generated after XhoI-SpeI digestion are indicated with their corresponding sizes in kb and were detected by Southern blot hybridization with a *LEU2* probe, depicted as a line with an asterisk. Note that the 2.9-kb band can also appear as a consequence of a break-induced replication event with the sister chromatid (sister chromatid BIR) or within the same chromatid (intrachromatid recombination, ICR), but both reactions are known to occur at a minor and irrelevant frequency. (b) Representative Southern blots and quantified repair ratios from time-course experiments performed at the indicated times after FLPm induction in wild type (WFLP), *rnh1Δ rnh201Δ* (WFR1R2) and *hpr1Δ* (WFHPR1) strains transformed with pTINV-FRT-T (FRT-T) or pTINV-FRT-NT (FRT-NT) under high transcription (n≥3). (c) Representative Southern blots and quantified repair ratios from time-course experiments performed at the indicated times after FLPm induction in wild type (WFLP), *rnh1Δ rnh201Δ* (WFR1R2) and *hpr1Δ* (WFHPR1) strains transformed with pTINV-FRT-T (FRT-T) or pTINV-FRT-NT (FRT-NT) under low transcription (n≥3). In (b, c), the 3.8-kb band corresponds to the intact plasmid, the 2.9-kb band to the repair intermediate, and 1.4 and 2.4-kb bands to the DSBs. Asterisks beside Southern blots indicate non-specific hybridization. Mean and SEM are plotted in (b, c) panels. In all cases, p>0.1 (two-way ANOVA test). See also *Figure 2—figure supplements 1* and *2*. Data underlying this figure are provided as *Figure 2—source data 1*. DSB, double-strand break; HR, homologous recombination; uSCE, unequal sister chromatid exchange; trx, transcription.

The online version of this article includes the following source data and figure supplement(s) for figure 2:

**Source data 1.** Effect of *rnh1Δ rnh201Δ* and *hpr1Δ* in the repair of replication-born DSBs with the sister chromatid.

**Figure supplement 1.** Analysis of FLPm-induced breaks and repair intermediates in *rnh1Δ rnh201Δ* and *hpr1Δ*.

**Figure supplement 1—source data 1.** Analysis of FLPm-induced breaks and repair intermediates in *rnh1Δ rnh201Δ* and *hpr1Δ*.

**Figure supplement 2.** Frequency of spontaneous and FLPm-induced recombination in *rnh1Δ rnh201Δ* and *hpr1Δ*.

**Figure supplement 2—source data 1.** Frequency of spontaneous and FLPm-induced recombination in *rnh1Δ rnh201Δ* and *hpr1Δ*.

systems. Analysis of Leu[+] events revealed that, as expected, spontaneous recombination levels were significantly higher in both *rnh1Δ rnh201Δ* and *hpr1Δ* cells (*Figure 2—figure supplement 2b*; *Aguilera and Klein, 1988*; *Amon and Koshland, 2016*; *Huertas and Aguilera, 2003*; *Stirling et al., 2012*; *Stuckey et al., 2015*). In contrast, FLPm-induced recombination did not decrease in any of the mutants but rather increased in *hpr1Δ* cells (*Figure 2—figure supplement 2c*). Thus, the impact of the *rnh1Δ rnh201Δ* and *hpr1Δ* mutations on the repair of replication-born DSBs by SCR was not detectable genetically in these systems, likely due to the fact that the high basal levels of Leu[+] events of the mutants that could mask a possible effect. In fact, this impact can be inferred from the fold increase of FLPm-induced recombination with respect to spontaneous levels, which was much lower in *rnh1Δ rnh201Δ* and *hpr1Δ* cells and particularly under high transcription conditions (*Figure 2—figure supplement 2d*). These results point to an impact of DNA-RNA hybrids in HR.

## DNA-RNA hybrid accumulation at sites undergoing DSBs

To assay whether DNA-RNA hybrids accumulate upon break induction in our repair systems, we performed DNA-RNA immunoprecipitation (DRIP)-qPCR experiments with the S9.6 antibody within the 255 bp region downstream of the break site before and after FLPm induction in the FRT-T and FRT-NT constructs (*Figure 3a*). The *hpr1Δ* mutation elevated the S9.6 signals by twofold regardless of the induction of FLPm expression. The high background levels of hybrids and damage originated in the S phase in *hpr1Δ* cells (*San Martin-Alonso et al., 2021*) likely masked any further increase. In contrast, *rnh1Δ rnh201Δ* caused a threefold increase specifically after FLPm induction (*Figure 3a*). This increase was transcription-dependent (*Figure 3b*) and was also observed when we analyzed the 317 bp region upstream of the break site (*Figure 3—figure supplement 1*). Importantly, all S9.6 signals were specific for DNA-RNA hybrids, since they were significantly reduced by in vitro RNase H treatment. As a control, we analyzed DNA-RNA hybrids at the R loop-prone *PDC1* gene in chromosome XII, which carries no *FRT* site, and detected no changes in hybrids regardless of whether FLPm was induced in any of the strains (*Figure 3c*). Thus, DNA-RNA hybrids accumulate at sites undergoing replication-born DSBs.

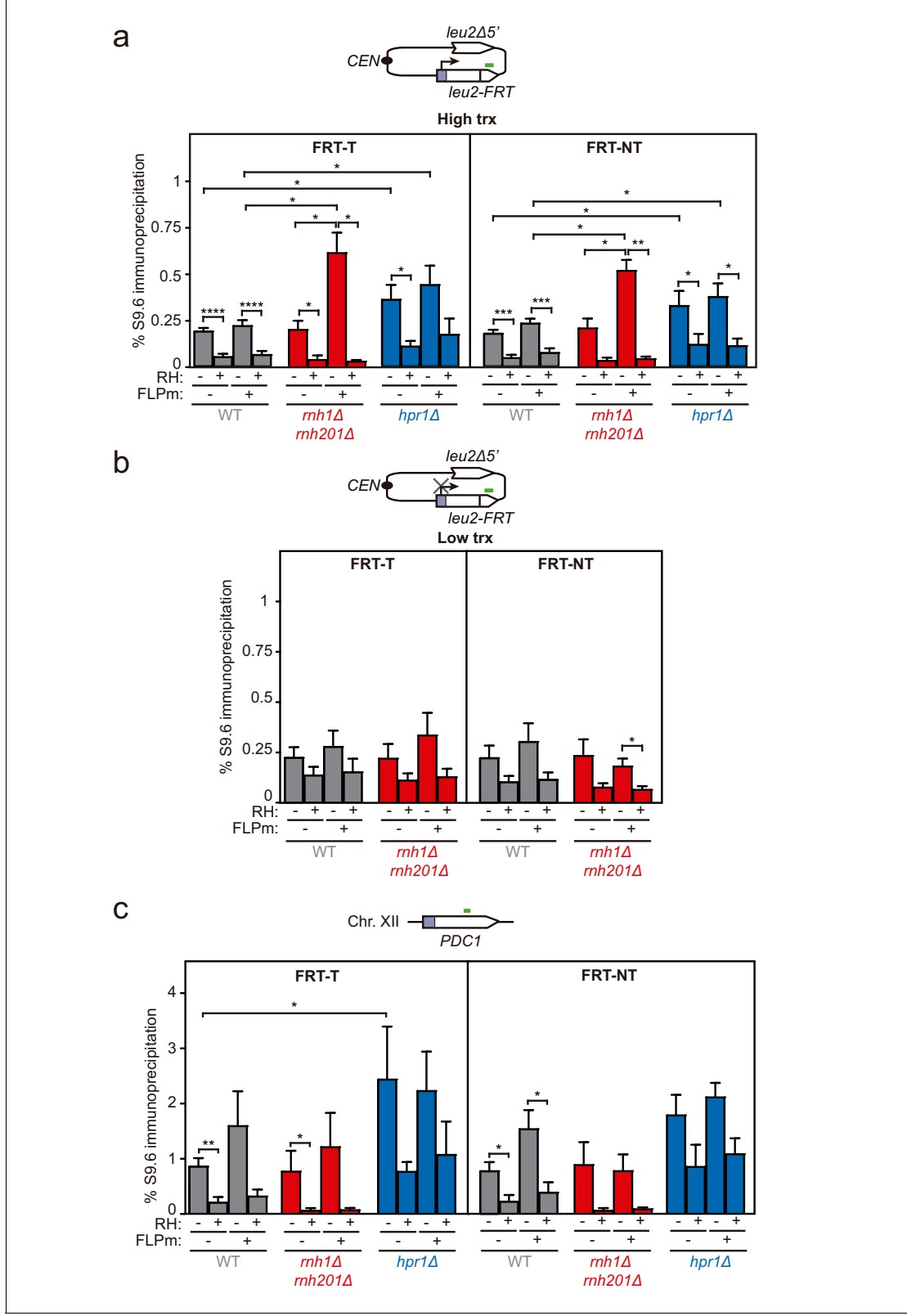

**Figure 3.** DNA-RNA hybrid accumulation at replication-born DSBs. (a) DRIP with the S9.6 antibody in the *leu2-FRT* alleles as depicted on top and in either spontaneous conditions (FLPm−) or after FLPm induction (FLPm+) in wild type (WFLP), *rnh1Δ rnh201Δ* (WFR1R2) and *hpr1Δ* (WFHPR1) strains transformed with pTINV-FRT-T (FRT-T) or pTINV-FRT-NT (FRT-NT) under high transcription and either non-treated (RH−) or after in vitro RNase H treatment (RH+) (n≥3). (b) DRIP with the S9.6 antibody in the *leu2-FRT* alleles as depicted on top and in either spontaneous conditions (FLPm−) or after

*Figure 3 continued on next page*

*Figure 3 continued*

FLPm induction (FLPm+) in wild type (WFLP) and *rnh1Δ rnh201Δ* (WFR1R2) strains transformed with pTINV-FRT-T (FRT-T) or pTINV-FRT-NT (FRT-NT) under low transcription and either non-treated (RH−) or after in vitro RNase H treatment (RH+) (n=4). (c) DRIP with the S9.6 antibody in the *PDC1* gene as depicted on top and in either spontaneous conditions (FLPm−) or after FLPm induction (FLPm+) in wild type (WFLP), *rnh1Δ rnh201Δ* (WFR1R2) and *hpr1Δ* (WFHPR1) strains transformed with pTINV-FRT-T (FRT-T) or pTINV-FRT-NT (FRT-NT) under high transcription and either non-treated (RH−) or after in vitro RNase H treatment (RH+) (n≥3). Mean and SEM are plotted in all panels. *p≤0.05; **p≤0.01; ***p≤0.001; ****p≤0.0001 (unpaired Student's t-test). See also *Figure 3—figure supplement 1*. Data underlying this figure are provided as *Figure 3—source data 1*. DRIP, DNA-RNA immunoprecipitation; trx, transcription.

The online version of this article includes the following source data and figure supplement(s) for figure 3:

**Source data 1.** DNA-RNA hybrid accumulation at replication-born DSBs.

**Figure supplement 1.** DNA-RNA hybrid accumulation upstream of the *FRT* site.

**Figure supplement 1—source data 1.** DNA-RNA hybrid accumulation upstream of the *FRT* site.

## DNA-RNA hybrids at DSBs directly generated by endonucleolytic cleavage impair repair

Next, to test whether hybrids were induced at DSBs formed directly by double-nucleolytic incision and how they influenced DSB repair, we constructed a new TINV system in which we introduced the full 117 bp *HO* site (TINV-FHO system) (*Figure 4a*). As in the previous systems, the 2.4- and 1.4-kb bands corresponded to DSBs and the 2.9-kb band to HR repair intermediates, which in this case would mostly result from BIR initiated from one of the DSB ends invading the truncated repeat located in the same chromatid, an intrachromatid recombination (ICR) reaction (*Figure 4a*). This is so because an intact sister chromatid to lead to an SCR event would only be present in G2 when only one of the chromatids was cleaved by chance (*Figure 4* and *Figure 4—figure supplement 1*). As expected, this *full-HO*-based TINV system yielded a much higher cleavage efficiency than the FRT-systems reaching up to 50% of DSBs (*Figure 4—figure supplement 1*). We noted a slightly faster repair under high transcription than under low transcription conditions, particularly at early time points (*Figure 4b and c*). Importantly, under high transcription, the ratio of repair molecules significantly decreased in both *rnh1Δ rnh201Δ* and *hpr1Δ* cells compared to wild type (*Figure 4b and d*). Furthermore, analysis of the frequency of Leu$^+$ recombinants (*Figure 5a*) revealed that both *rnh1Δ rnh201Δ* and *hpr1Δ* mutants elevated spontaneous recombination as expected (*Figure 5—figure supplement 1a*), but led to a significant decrease in HO-induced recombination specifically under high transcription of the cleaved region (*Figure 5b*). Notably, the reduction in HO-induced recombination frequency was partially suppressed when RNase H1 overexpression (*Figure 5c*), implying that DNA-RNA hybrids impair the formation of the HR products, consistent with the view that hybrids at DSB sites impair HR repair.

We then confirmed the accumulation of DNA-RNA hybrids by DRIP-qPCR at the DSBs in this TINV-FHO system. A fourfold increase in the S9.6 signal was detected upstream of the break site upon DSB-induction already in wild-type cells (*Figure 5d*). Provided that *hpr1Δ* increases background hybrids making it difficult to see a further increase mediated by DNA breaks, as shown in *Figure 3a*, we just used *rnh1Δ rnh201Δ* mutants to test whether, when not removed, hybrids could be seen accumulated at higher levels at DSBs. Notably, hybrids significantly increased in *rnh1Δ rnh201Δ* cells (*Figure 5d*). Again, a similar increase was observed downstream of the break site (*Figure 5—figure supplement 1b*), implying that hybrids accumulate at both sites of the break. Such break-induced S9.6 signals were partially dependent on transcription (*Figure 5e*) and were not observed at the *PDC1* gene (*Figure 5f*).

Altogether, these results indicate that high levels of DNA-RNA hybrids formed at endonuclease-induced breaks negatively interfere with DSB repair.

## DNA-RNA hybrids interfere with the repair of chromosomal DSBs

Although plasmid systems have been recurrently validated as models to study DNA repair and recombination, we wanted to confirm our conclusions in chromosomal DSBs to make sure that any putative local difference in chromatin or topology, even though unlikely, did not affect results. For this, we developed an allelic recombination system (DGL-FRT), consisting in a diploid yeast strain carrying two versions of the *lys2* allele in each of the homologous chromosomes II (*Figure 6a*). One

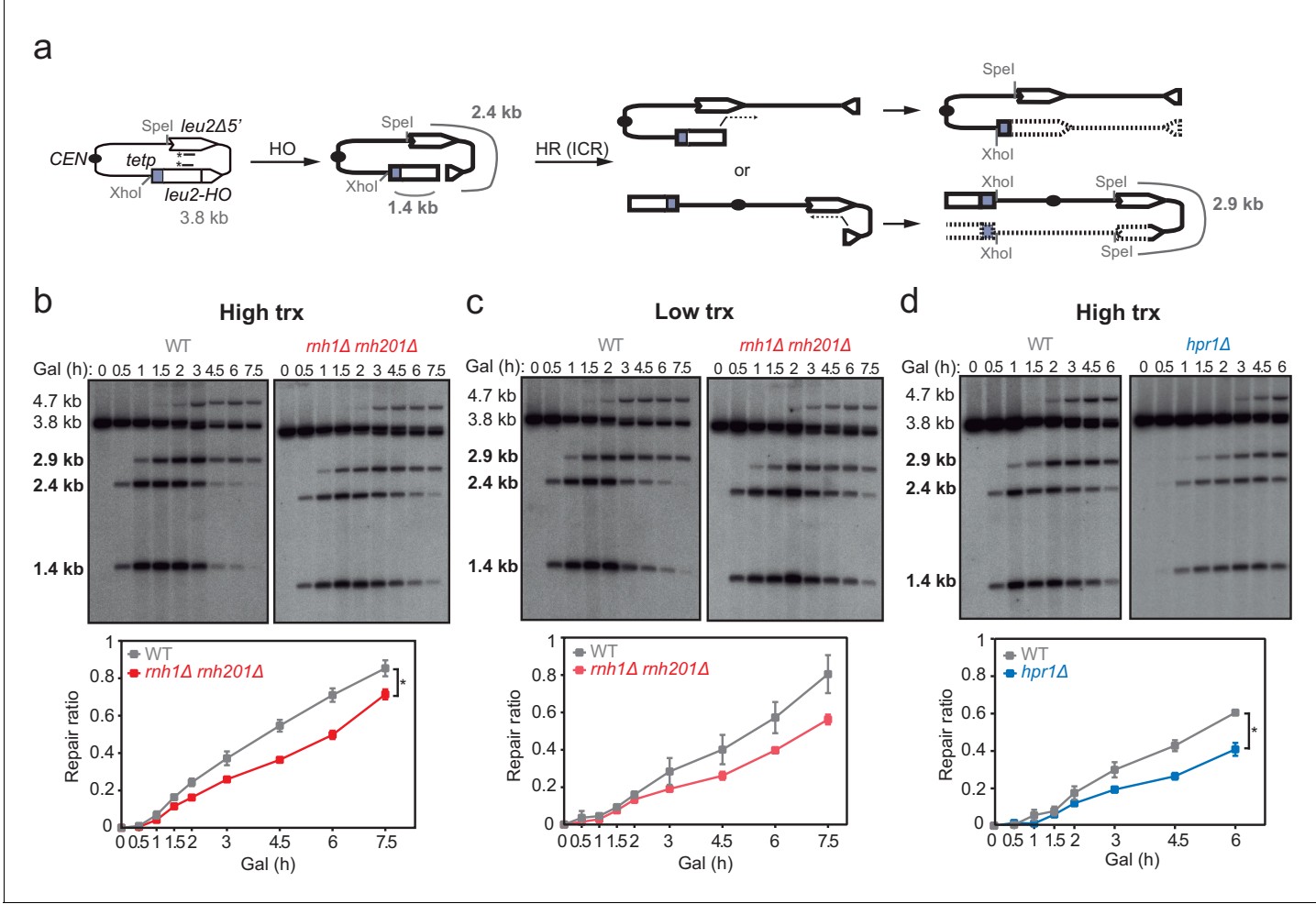

**Figure 4.** Effect of *rnh1Δ rnh201Δ* and *hpr1Δ* in the repair of endonuclease-induced DSBs. (**a**) Schemes of the TINV-FHO system, in which HO induction leads to replication-independent DSBs which when occurring by break-induced replication (BIR) from one of the DSB ends invading the truncated repeat located in the same chromatid (intrachromatid recombination, ICR) leads to the depicted repair intermediate. Other recombination reactions (such as uSCE depicted in *Figure 2*) are also possible. Fragments generated after XhoI-SpeI digestion are indicated with their corresponding sizes in kb and were detected by Southern blot hybridization with a *LEU2* probe, depicted as a line with an asterisk. (**b**) Representative Southern blots and quantified repair ratios from time-course experiments performed at the indicated times after HO induction in wild type (WS) and *rnh1Δ rnh201Δ* (WSR1R2) strains transformed with pTINV-FHO under high transcription (n=4). (**c**) Representative Southern blots and quantified repair ratios from during time-course experiments performed at the indicated times after HO induction in wild type (WS) and *rnh1Δ rnh201Δ* (WSR1R2) strains transformed with pTINV-FHO under low transcription (n=3). (**d**) Representative Southern blots and quantified repair ratios from time-course experiments performed at the indicated times after HO induction in wild type (WS) and *hpr1Δ* (WSHPR1) strains transformed with pTINV-FHO under high transcription (n=3). In (**b–d**), the 3.8-kb band corresponds to the intact plasmid, 2.9-kb band to the repair intermediates, and 1.4 and 2.4-kb bands to the DSBs. The 4.7-kb band corresponds to a repair intermediate that has not been used for the quantification analysis. Mean and SEM are plotted in (**b–d**) panels. *p≤0.05 (two-way ANOVA test). See also *Figure 4—figure supplement 1*. Data underlying this figure are provided as *Figure 4—source data 1*. DSB, double-strand break; HR, homologous recombination; ICR, intrachromatid recombination; trx, transcription.

The online version of this article includes the following source data and figure supplement(s) for figure 4:

**Source data 1.** Effect of *rnh1Δ rnh201Δ* and *hpr1Δ* in the repair of endonuclease-induced DSBs.

**Figure supplement 1.** Analysis of HO-induced DSBs and repair intermediates in *rnh1Δ rnh201Δ* and *hpr1Δ*.

**Figure supplement 1—source data 1.** Analysis of HO-induced DSBs and repair intermediates in *rnh1Δ rnh201Δ* and *hpr1Δ*.

of the *lys2* alleles was under the control of the *GAL1* promoter (*GALp*) and contained the *FRT* site in either FRT-T or FRT-NT orientation, and the other carried a 1 bp deletion at position 3705 (*lys2-3705*). In this system, replication-born DSBs at each of the FRT sites would lead to Lys$^+$ recombinants by allelic recombination between the homologous chromosomes. Spontaneous recombination frequencies were below $10^{-4}$ and similar in both constructs and both transcriptional conditions

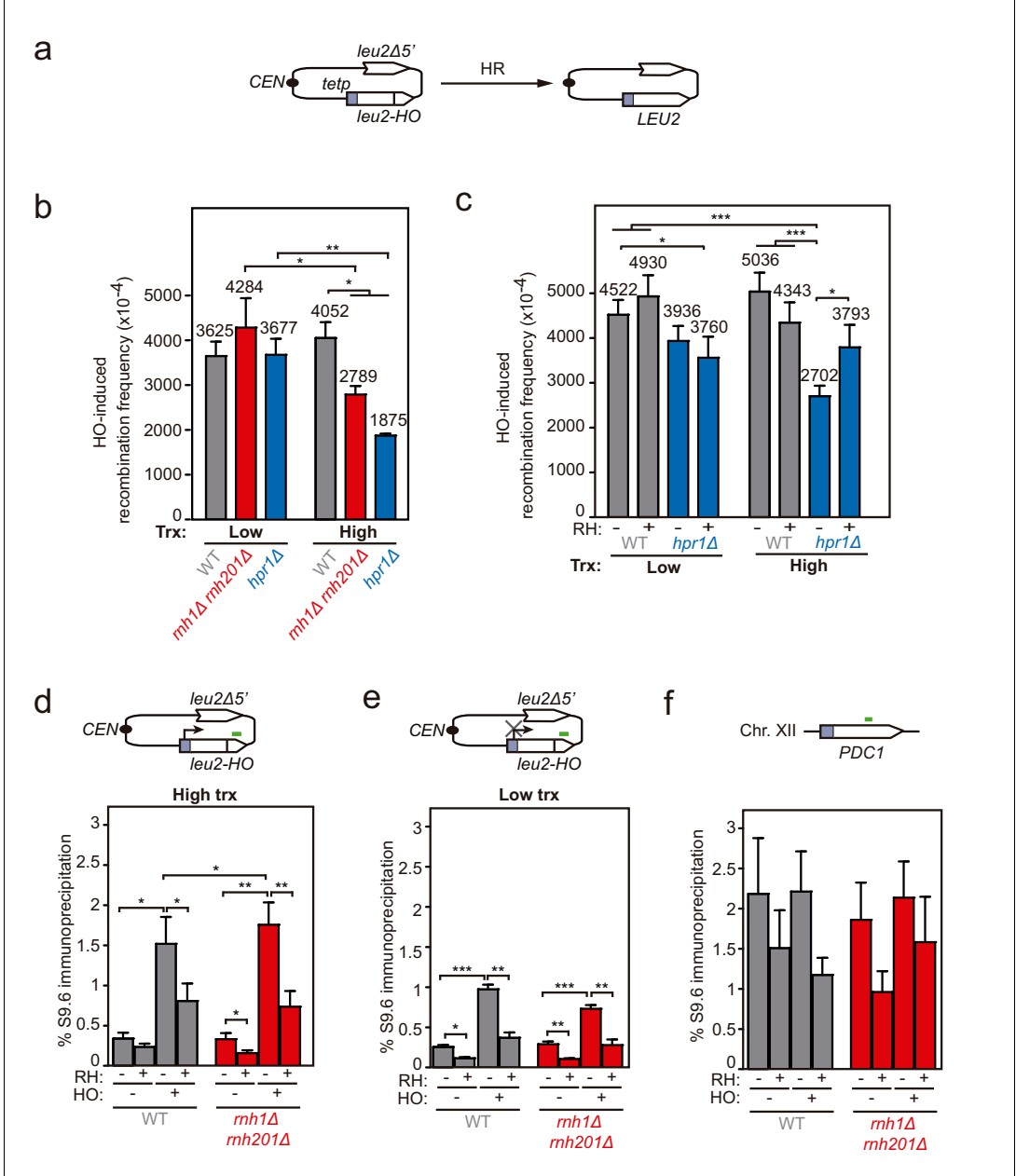

**Figure 5.** Genetic analysis of the repair and DNA-RNA hybrid accumulation at endonuclease-induced DSBs. (a) Scheme of the pCM189-L2FHO recombination system. (b) Frequency of HO-induced recombination in wild type (WS), *rnh1Δ rnh201Δ* (WSR1R2) and *hpr1Δ* (WSHPR1) strains transformed with pTINV-FHO under low or high transcription (n≥3). (c) Frequency of HO-induced recombination in wild type (WS) and *hpr1Δ* (WSHPR1) strains transformed with either pRS314 (RH−) or pRS314-GALRNH1 (RH+) and pTINV-FHO under low or high transcription (n=9). (d) DRIP with the S9.6 antibody in the *leu2-HO* allele as depicted on top and in either spontaneous conditions (HO−) or after HO induction (HO+) in wild type (WS) and *rnh1Δ rnh201Δ* (WSR1R2) strains transformed with pTINV-FHO under high transcription and either non-treated (RH−) or after in vitro RNase H treatment (RH+) (n=5). (e) DRIP with the S9.6 antibody in the *leu2-HO* allele as depicted on top and in either spontaneous conditions (HO−) or after HO induction (HO+) in wild type (WS) and *rnh1Δ rnh201Δ* (WSR1R2) strains transformed with pTINV-FHO under low transcription and either non-treated (RH−) or after in vitro RNase H treatment (RH+) (n=4). (f) DRIP with the S9.6 antibody in the *PDC1* gene as depicted on top and in either spontaneous conditions (HO−) or after HO induction (HO+) in wild type (WS) and *rnh1Δ rnh201Δ* (WSR1R2) strains transformed with pTINV-FHO under high transcription and either untreated (RH−) or after in vitro RNase H treatment (RH+) (n=5). Mean and SEM of independent experiments consisting in the median value of six independent colonies each are plotted in (b–f) panels. *p≤0.05; **p≤0.01; ***p≤0.001 (unpaired Student's t-test in (b) panel and paired Student's t-test in (c–f) panels). See also *Figure 5—figure supplement 1*. Data underlying this figure are provided as *Figure 5—source data 1*. DRIP, DNA-RNA immunoprecipitation; HR, homologous recombination; trx, transcription.

The online version of this article includes the following source data and figure supplement(s) for figure 5:

*Figure 5 continued on next page*

*Figure 5 continued*

**Source data 1.** Genetic analysis of the repair and DNA-RNA hybrid accumulation at endonuclease-induced DSBs.
**Figure supplement 1.** Frequency of recombination and hybrid accumulation upstream of the *HO* site.
**Figure supplement 1—source data 1.** Frequency of recombination and hybrid accumulation upstream of the *HO* site.

(*Figure 6b*). As expected, induction of the FLPm nickase boosted recombination up to $10^{-1}$ (*Figure 6c*). Interestingly, FLP-induced recombination was 2.3-fold lower under high transcription and in an RNase H1-sensitive manner in both FRT-T and FRT-NT constructs (*Figure 6c*). These results suggest that DNA-RNA hybrids also interfere with the repair of DSBs occurring in chromosomes. In support of these conclusions, we confirmed by DRIP-qPCR that indeed DNA-RNA hybrids accumulated, as tested within the 81 bp region upstream and the 128 bp region downstream of the DSB (*Figure 6—figure supplement 1* and *Figure 6d*). Again, hybrids were not detected under low transcription (*Figure 6e*) or at the *PDC1* locus, used as an FRT-free control (*Figure 6f*). Hence, we can conclude that DNA-RNA hybrids accumulate at DNA breaks and interfere with their repair by HR.

## Discussion

In this work, we show that DNA-RNA hybrids accumulate upon DSB induction in transcribed DNA. This phenomenon happens regardless of the origin of the break, whether replication-born or direct endonucleolytic cleavage, and whether in plasmid-born or chromosomal recombination systems. Importantly, rather than helping DNA repair, they can interfere with the repair by HR causing genetic instability, which suggests that hybrids at DNA breaks are mainly the result of the accidental co-transcriptional event facilitated by the release of the supercoil constraint.

We detected DNA-RNA hybrids at the break site even in wild-type cells (*Figures 5* and *6*). This seems to happen when the efficiency of break induction is high enough (*Figure 5*) since, upon low cleavage induction, such as in the case of the FLPm-induced DSBs, it was necessary to delete the DNA-RNA hybrid resolution machinery, such as the RNases H, to observe such break-induced hybrids (*Figure 3*). This result supports the previous reports showing that DNA-RNA hybrid removal by RNase H enzymes contributes to DSB repair (*Amon and Koshland, 2016*; *Britton et al., 2014*; *Ohle et al., 2016*) in addition to removal by helicases, as shown for Senataxin, DDX1, or DDX5 in human cells (*Cohen et al., 2018*; *Li et al., 2016*; *Sessa et al., 2021*; *Yu et al., 2020*) and possibly other factors yet to be explored. Indeed, persistent hybrids caused by depletion of RNase H, helicases, or the human exosome have been shown to affect RPA binding and/or DNA end resection in yeast or human cell studies (*Daley et al., 2020*; *Domingo-Prim et al., 2019*; *Li et al., 2016*; *Ohle et al., 2016*; *Rawal et al., 2020*; *Sessa et al., 2021*; *Yu et al., 2020*). It is therefore possible that DNA-RNA hybrid accumulation in *hpr1Δ* and *rnh1Δ rnh201Δ* affects repair by negatively interfering with DSB resection. This could be particularly important when the hybrid covers the 5′ end that needs to be resected, although we have observed the formation of break-induced DNA-RNA hybrids at both sides of the break (*Figure 3*, *Figure 3—figure supplements 1*, *Figure 5*, *Figure 5—figure supplements 1*, *Figure 6* and *Figure 6—figure supplement 1*). The hybrid formed in the 3′ end might initially favor resection of the complementary strand as we previously proposed (*Aguilera and Gómez-González, 2017*), but we envision that it would need to be removed later to allow the loading of RPA and subsequently Rad51. The increased loss of transcriptionally active cleaved plasmids in DNA-RNA hybrid-accumulating mutants (*Figure 1*) may be explained as a consequence of the observed HR defects. Similarly, DNA-RNA hybrids accumulated at breaks in Senataxin-depleted cells have been shown to channel repair towards NHEJ with deleterious consequences such as increased translocations in yeast and lethality in human cells (*Cohen et al., 2018*; *Rawal et al., 2020*).

The detection of break-induced DNA-RNA hybrids at both sides of the break (*Figure 3*, *Figure 3—figure supplement 1*, *Figure 5*, *Figure 5—figure supplement 1*, *Figure 6* and *Figure 6—figure supplement 1*) is in agreement with the results of a recent report in which DNA-RNA hybrids were detected at both sides of HO cleavage at the MAT loci, particularly upon Senataxin depletion (*Rawal et al., 2020*). However, we show that this phenomenon is completely dependent on

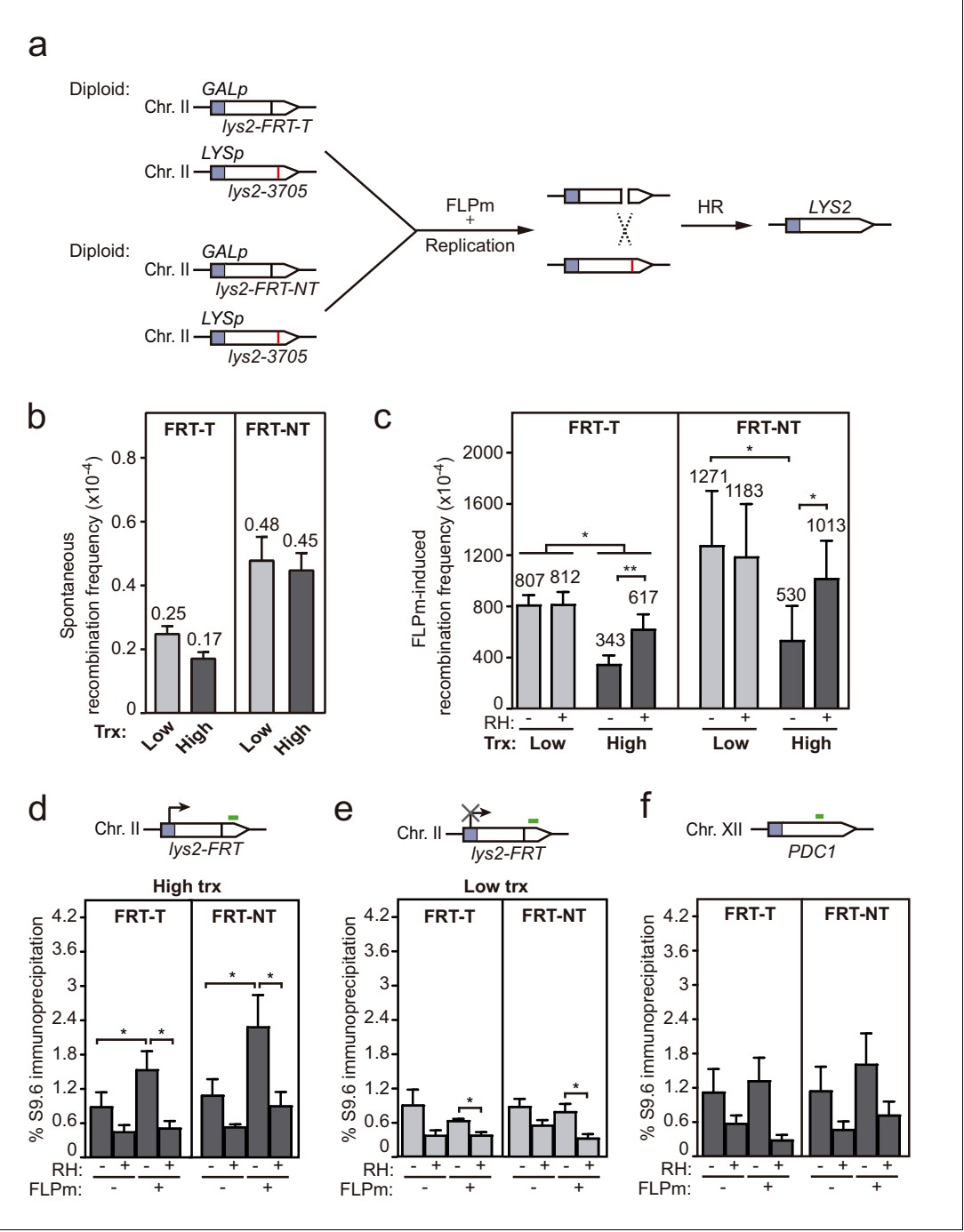

**Figure 6.** Interference of DNA-RNA hybrids with DSB repair in a chromosome. (a) Scheme of the diploid chromosome-based FLPm recombination systems (DGL-FRT-T and NT), in which FLPm induction of nicks leads to replication-born DSBs that when repaired with the homologous chromosome would lead to the restoration of the *LYS2* gene. (b) Frequency of spontaneous recombination in DGLFT and DGFLNT strains carrying the FRT-T and NT constructs respectively and transformed with pCM190 under low or high transcription (n=3). (c) Frequency of FLPm-induced recombination DGLFT and DGFLNT strains carrying the FRT-T and NT constructs respectively and transformed with pCM190-FLP and either pRS313 (RH−) or pRS313-GALRNH1 (RH+) under low or high transcription (n=5). (d) DRIP with the S9.6 antibody in the *lys2-FRT* alleles as depicted on top and in either spontaneous conditions (FLPm−) or after FLPm induction (FLPm+) in GLFT and GFLNT strains carrying the FRT-T and NT constructs respectively, transformed with pCM190-FLP under high transcription and either non-treated (RH−) or after in vitro RNase H treatment (RH+) (n=4). (e) DRIP with the S9.6 antibody in the *lys2-FRT* alleles as
*Figure 6 continued on next page*

*Figure 6 continued*
depicted on top and in either spontaneous conditions (FLPm−) or after FLPm induction (FLPm+) in GLFT and GFLNT strains carrying the FRT-T and NT constructs, respectively, transformed with pCM190-FLP under low transcription and either non-treated (RH−) or after in vitro RNase H treatment (RH+) (n=4). (f) DRIP with the S9.6 antibody in the *PDC1* gene as depicted on top and in either spontaneous conditions (FLPm−) or after FLPm induction (FLPm+) in GLFT and GFLNT strains carrying the FRT-T and NT constructs, respectively, transformed with pCM190-FLP under high transcription and either non-treated (RH−) or after in vitro RNase H treatment (RH+) (n=3). Mean and SEM of independent experiments consisting in the median value of six independent colonies each are plotted in (**b–f**) panels. *p≤0.05; **p≤0.01 (paired Student's t-test). See also *Figure 6—figure supplement 1*. Data underlying this figure are provided as *Figure 6—source data 1*. DRIP, DNA-RNA immunoprecipitation; DSB, double-strand break; HR, homologous recombination; trx, transcription.
The online version of this article includes the following source data and figure supplement(s) for figure 6:

**Source data 1.** Interference of DNA-RNA hybrids with DSB repair in a chromosome.
**Figure supplement 1.** DNA-RNA hybrids accumulation upstream the *FRT* site.
**Figure supplement 1—source data 1.** DNA-RNA hybrids accumulation upstream the *FRT* site.

transcription of the construct driven by the *tetp* in the case of FLPm-induced breaks (*Figures 3* and *6*) and partially dependent in the case of HO-induced breaks (*Figure 5*). Therefore, we conclude that DNA breaks lead to DNA-RNA hybrids at both sides of the break due to pre-existing ongoing transcription rather than the de novo RNA synthesis previously proposed (*Ohle et al., 2016*; *Rawal et al., 2020*). In agreement with our observations, analysis of DSBs-induced genome-wide in human cell cultures has recently shown that pre-existing transcription is critical for the formation of DNA-RNA hybrids at breaks (*Bader and Bushell, 2020*; *Cohen et al., 2018*). Based on the observation that R-loops are induced in mutants with increased RNA polymerase II backtracking in human cells (*Zatreanu et al., 2019*), it has been proposed that RNA polymerase backtracking could be the source of hybrids upstream of the break site (*Marnef and Legube, 2021*). However, we cannot discern from the DRIP analysis whether hybrids at both sides of the break are formed by different or the same RNA molecule. In any case, we favor the idea that break-induced hybrids are rather the incidental consequence of DNA breakage during transcription (*Figure 7*). Indeed, the transient transcriptional shutdown that is known to happen soon after DSBs promoting repair (*Pankotai et al., 2012*; *Shanbhag et al., 2010*) might also contribute to preventing such incidental hybridization. Our results therefore disfavor the possibility that such DNA-RNA hybrids are an intermediate required for the repair reaction in contrast to other studies (*D'Alessandro et al., 2018*; *Keskin et al., 2014*; *Liu et al., 2021*; *Lu et al., 2018*; *Ohle et al., 2016*; *Ouyang et al., 2021*). Indeed, RNase H overexpression caused no defect in our repair systems (*Figure 5*).

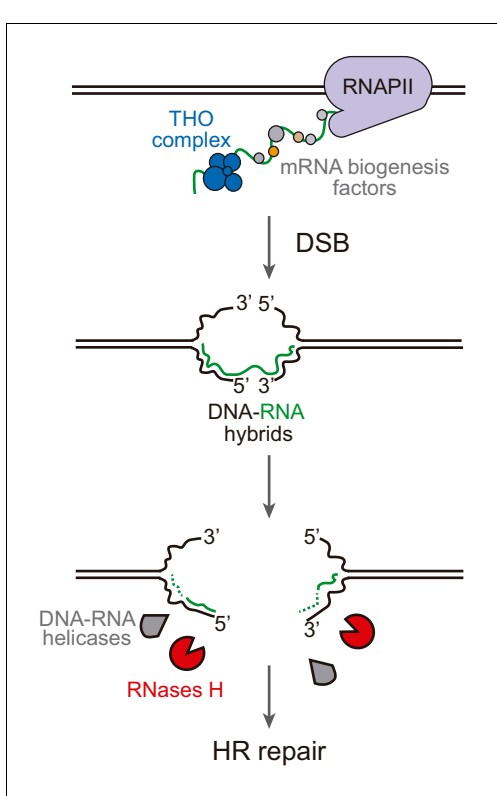

**Figure 7.** Model of DNA-RNA hybrid formation upon breakage of transcribed DNA. During transcription, nascent RNA is coated by mRNA biogenesis factors, such as the THO complex, that prevent RNA from hybridizing with its complementary DNA. In their absence, DNA-RNA hybrids can remain even when the DNA is broken. Moreover, double-strand break (DSB) induction leads to incidental DNA-RNA hybridization at both sides of the break site. Such hybrids need to be removed by RNase H enzymes or helicases to allow further repair by homologous recombination (HR).

A nice report in human cells has recently shown that transcription increases HR frequency after break induction (*Ouyang et al., 2021*), which supported by in vitro studies with synthetic DNA-RNA hybrids and Cas9-mediated DNA-RNA hybrid formation led to the intriguing proposal that hybrids are regular intermediates of HR with a positive role in the reaction (*Ouyang et al., 2021*). However, we did not observe that transcription, required to generate the RNA involved in hybrids, increased the frequency of HR after break induction in any of our systems (see *Figure 2—figure supplement 2c*, *Figure 5b*, and *Figure 6b*). Indeed, it has been shown that transcription can stimulate HR just as a consequence of the chromatin status of transcriptionally active loci (*Aymard et al., 2014*; *Clouaire and Legube, 2015*). This, together with the fact that we clearly detected break-induced hybrids in a transcription-dependent manner (*Figures 3, 5* and *6*) but no positive effect of transcription on HR frequencies (*Figure 5*, *Figure 6* and *Figure 2—figure supplement 2*) suggests that DNA-RNA hybrids rather than being an intermediate with a positive role in DSB repair, forms accidentally at breaks interfering with their repair. Certainly, we cannot discard that hybrids formed at specific DNA regions in a regulated manner or formed by a Cas9-driven reaction could play a positive role in DSB repair. Notwithstanding, it is also worth mentioning that the scenario we propose is compatible with the scheduled synthesis of damage-induced ncRNAs de novo, which could serve the purpose of DSB signaling (*Burger et al., 2019*; *D'Alessandro et al., 2018*; *Francia et al., 2012*; *Michelini et al., 2017*; *Pessina et al., 2019*; *Vítor et al., 2019*).

Our systems allow us to conclude that break-induced hybrids are not only transcription-dependent but independent of the origin of the DSB, which we induced by either direct endonucleolytic HO cleavage (*Figure 5*) or by the replication of single-strand breaks (SSBs) caused by FLPm at the T or NT strands (*Figure 3*). Although directing the SSBs to the T or NT strand led to DSBs that were repaired with the same efficiency (*Figure 2*), we observed that the appearance of repair intermediates was slightly faster when transcription was high than when transcription was low (*Figure 4*). This has been reported before (*Chaurasia et al., 2012*) and could be attributed to increased accessibility of repair factors to transcribed regions, as recently shown for human RAD51 and RAD52 (*Aymard et al., 2014*; *Wei et al., 2015*; *Yasuhara et al., 2018*). Thus, the repair of transcribed DNA might be at the same time hampered by the incidental formation of hybrids, as we observed here, and fostered by the enhanced recruitment of the repair machinery.

A priori, all conditions promoting DNA-RNA hybridization (break-specific or not) could potentially impair HR at transcribed regions since HR was affected even when DNA-RNA hybrids were already accumulated before cleavage induction, as it is the case of the *hpr1Δ* mutant from the THO complex (*Figure 4*). Therefore, our results support a model (*Figure 7*) in which DNA-RNA hybrids at DSBs, either pre-existing or promoted by the induction of the break at transcribed loci, need to be removed in order to allow further repair and maintain genome stability. Hybrid removal at breaks could potentially be performed by multiple redundant factors in addition to RNases H. However, not all factors that have been shown to remove DNA-RNA hybrids must necessarily act at breaks. We envision that both overlapping and specific functions counteract the harmful potential of DNA-RNA hybrids in each physiological process affected. On the one hand, a cohort of factors has evolved to prevent or remove DNA-RNA hybrids co-transcriptionally, these factors being likely associated with transcription elongation as exemplified by the THO complex and its interaction with the Sub2/UAP56 helicase (*Pérez-Calero et al., 2020*). On the other hand, hybrids can be dissolved during replication and, when causing replication fork impairments, they are counteracted by replication-associated repair factors, such as the Fanconi anemia pathway (*García-Rubio et al., 2015*; *Hatchi et al., 2015*; *Schwab et al., 2015*) or the SWI/SNF chromatin remodeling complex (*Bayona-Feliu et al., 2021*). Similarly, hybrids hampering DSB repair can be counteracted by DSB repair-associated factors, as exemplified in human cells by BRCA2 retaining the DDX5 helicase at DSBs boosting its activity to unwind DNA-RNA hybrids (*Bhatia et al., 2014*; *Mersaoui et al., 2019*; *Sessa et al., 2021*) or in yeast by the recruitment of Senataxin to DSBs by Mre11 (*Rawal et al., 2020*). Further research would be required to unravel the final puzzle of how DNA-RNA hybrids are physiologically regulated in each of the circumstances and which general and specific DNA-RNA hybrid counteracting factors have a function at DNA breaks.

# Materials and methods

**Key resources table**

| Reagent type (species) or resource | Designation | Source or reference | Identifiers | Additional information |
|---|---|---|---|---|
| Genetic reagent *Saccharomyces cerevisiae* | W303 background strains with different gene deletions | Various | | (See Materials and methods section) |
| Recombinant DNA reagent | Yeast expression plasmids and recombination systems | Various | | (See Materials and methods section) |
| Sequence-based reagent | Primers for DRIP, RT-PCR and probe amplification | Condalab | | (See Materials and methods section) |
| Antibody | S9.6 anti DNA:RNA hybrids (mouse monoclonal) | ATCC Hybridoma cell line | Cat # HB-8730, RRID:CVCL_G144 | (1 mg/ml) |
| Commercial assay kit | Macherey-Nagel DNA purification | Macherey-Nagel | Cat # 740588.250 | |
| Commercial assay kit | Qiagen's RNeasy | Qiagen | Cat # 75162 | |
| Commercial assay kit | Reverse transcription kit | Qiagen | Cat # 205311 | |
| Peptide, recombinant protein | Zymolyase 20T | US Biological | Z1001 | (15 mg/ml) |
| Chemical compound, drug | Doxycyclin hyclate | Sigma-Aldrich | D9891 | (5 mg/ml) |
| Peptide, recombinant protein | Proteinase K (PCR grade) | Roche | Cat # 03508811103 | |
| Peptide, recombinant protein | Rnase A | Roche | Cat # 10154105103 | |
| Peptide, recombinant protein | Rnase III | Thermo Fisher Scientific | Cat # AM2290 | |
| Peptide, recombinant protein | Spermidine | Sigma-Aldrich | Cat # S2626 | |
| Peptide, recombinant protein | Spermine | Sigma-Aldrich | Cat # S3256 | |
| Other | iTaq Universal SYBR Green | Bio-Rad | Cat # 1725120 | |
| Software, algorithm | GraphPad Prism V8.4.2 | GraphPad Software, La Jolla, CA, USA | RRID:SCR_002798 | |

## Yeast strains and plasmids

Yeast strains and plasmids used in this study are listed and described in *Supplementary file 1*.

pCM189-L2FRT-T and pCM189-L2FRT-NT, carrying the *leu2-FRT-T* and *leu2-FRT-NT* alleles, were constructed by cloning the BamHI-HindIII fragment of pRS316-FRTa and pRS316-FRTb (*Ortega et al., 2019*), respectively, into BamHI-HindIII digested pCM189-L2HOr (*González-Barrera et al., 2003*). Note that the *leu2-FRT-T* allele was previously published as *leu2-FRT* (*Ortega et al., 2019*) but has been re-named here for clarification.

pRS316-FHO was generated by cloning the EcoRI-digested 117-bp HO sequence, which was previously obtained by gene synthesis (gBlocks Gene Fragments, IDT), into EcoRI-digested pRS316-LEU2 (*Ortega et al., 2019*). pTINV-FRT-T and pTINV-FRT-NT plasmids were previously described as pTINV-FRT and pTINV-FRTb (*Ortega et al., 2019*) but were re-named here for clarification. pTINV-FHO was constructed by cloning the BstEII-HindIII fragment of pRS316-FHO into BamHI-HindIII digested pTINV-HO (*González-Barrera et al., 2003*). pCM189-L2FHO was constructed by cloning the BamHI-HindIII fragment of pRS316-FHO into BamHI-HindIII digested pCM189-L2HOr (*González-Barrera et al., 2002*). pCM190-FLP was constructed by cloning the BamHI-digested FLPm fragment, obtained by PCR amplification of pBIS-GALkFLP (*Tsalik and Gartenberg, 1998*) with primers FLP_BamHI_Fw and FLP_Rv (*Supplementary file 2*), into pCM190 (*Garí et al., 1997*). pRS313 was previously described (*Sikorski and Hieter, 1989*). pRS314-GALRNH1 was constructed by cloning the SalI-SpeI fragment from pRS313-GALRNH1 (*García-Benítez et al., 2017*) into pRS314 (*Sikorski and Hieter, 1989*).

GLY strain was generated by replacement of the *LYS2* promoter with the *NATNT2::GAL* fragment, which was amplified by PCR from a pFA6aNATNT2-GAL plasmid derived from pFA6aNATNT2 (*Janke et al., 2004*).

GLFT and GLFNT strains containing the FRT sequence at position 2952 of the *LYS2* gene were generated by transformation of the GLY strain with a PCR product amplified from pTINV-FRT (*Ortega et al., 2019*) with primers LYSFRTT_Fw and LYSFRTT_Rv or LYSFRTNT_Fw and LYSFRTNT_Rv (*Supplementary file 2*), together with the pML104-LYS2g plasmid to express Cas9 and a 20mer guide (TACATCCTTGCAGATTTGTT). pML104-LYS2g plasmid resulted from the insertion of an *LYS2* region (from nucleotide 2953 to 2972), which was obtained by primer annealing (LYS2_2953-72_Fw and LYS2_2953-72_Rv) (*Supplementary file 2*), into BclI-SwaI-digested pML104 (*Laughery et al., 2015*).

YLY strain was generated by inducing a single-bp deletion at position 3705 of the *LYS2* gene of the YBP250 wild-type strain using the pML104-3′mut plasmid, which contains the Cas9 and a 20mer guide sequence (GCCAATTCATTTTCTTTGGG). pML104-3′mut plasmid was constructed by inserting the *LYS2* region from nucleotide 3700 to 3719, which was obtained by primer annealing (LYS2_3700-3719_Fw and LYS2_3700-3719_Rv) (*Supplementary file 2*), into BclI-SwaI-digested pML104 (*Laughery et al., 2015*).

DGLFT and DGLFNT strains were generated by crossing the YLY strain with GLFT and GLFNT strains, respectively.

## DRIP

In either spontaneous conditions or after 3 hr of HO or FLP induction in the case of the TINV-FRT and TINV-FHO systems and 5.5 hr in the case of the GL-FRT system, DRIP was performed essentially as previously described (*García-Rubio et al., 2018*). Briefly, cultures were collected, washed two times with cold water, resuspended in 1.2 ml spheroplasting buffer (1 M sorbitol, 10 mM EDTA pH 8, 0.1% β-mercaptoethanol, and 2 mg/ml Zymolyase 20T) and incubated 35 min at 30°C to obtain spheroplasts. Pellets were resuspended in 565 μl buffer G2 (800 mM Guanidine HCl, 30 mM Tris-Cl pH 8, 30 mM EDTA pH 8, 5% Tween-20, and 0.5% Triton X-100) and treated with 50 μl RNase A (10 mg/ml, Roche) for 90 min at 37°C and 80 μl of proteinase K (20 mg/ml, Roche) for 120 min at 50°C. Cell debris was eliminated by centrifugation and DNA was extracted with chloroform:isoamyl alcohol (24:1) and isopropanol. DNA was collected with a glass Pasteur pipette, washed with 70% EtOH, resuspended in 1× TE and digested overnight with 50U HindIII, HincII, BsrGI, AflII, SspI (New England Biolabs), and 2.5U Rnase III (Thermo Fisher Scientific). Half of the DNA was treated with 60U RNase H (New England Biolabs) overnight at 37°C. Immunoprecipitation using Dynabeads Protein A (Thermo Fisher Scientific) for S9.6 monoclonal antibody (10 mg/ml final concentration, hybridoma cell line HB-8730) was carried out at 4°C for 180 min in 500 μl 1× binding buffer (10 mM NaPO$_4$ pH 7.0, 140 mM NaCl, and 0.05% Triton X-100) and samples were washed three times with 1× binding buffer. Chromatin was eluted at 55°C for 45 min in 100 μl elution buffer (50 mM Tris pH 8.0, 10 mM EDTA, and 0.5% SDS) with 7 μl proteinase K (20 mg/ml). DNA was cleaned up with a Macherey-Nagel purification kit. Real-time quantitative PCR was performed using iTaq universal SYBR Green (Bio-Rad) with a 7500 Real-Time PCR machine (Applied Biosystems). The PCR primers used were FRT_Fw, FRT_Rv, PDC1_Fw, PDC1_Rv, 5FRT_Fw, 5FRT_Rv, 3K_Fw, 3K_Rv, 2.5K_Fw, and 2.5K_Rv

(*Supplementary file 2*). The mean value of the % of input of at least three independent transformants was plotted but numerical data can be seen in the source data file.

## Quantification of mRNA levels

RNA was extracted with Qiagen's RNeasy kit. cDNA synthesis was performed with QuantiTect Reverse Transcription kit (Qiagen). *leu2* mRNA relative levels were obtained using FRT_Fw, FRT_Rv, ACT1_Fw, and ACT1_RV primers (*Supplementary file 2*). Relative mRNA levels were calculated normalizing the data from cultures in high (SRaf media) versus low (SRaf media with 5 µg/ml doxycycline) transcription conditions for each transformant. Numerical data can be seen in the source data file.

## Physical analysis of HR intermediates

DNA was extracted from each collected sample as previously described (*Gómez-González et al., 2011*). Briefly, cultures were collected, washed two times with cold water, resuspended in 400 µl NIB (17% (w/v) glycerol, 50 mM (3-[Nmorpholino] propanesulfonic acid) sodium salt (MOPS, Sigma-Aldrich) pH 7.5, 150 mM $CH_3CO_2K$ (Sigma-Aldrich), 2 mM $MgCl_2$, 500 µM spermidine (Sigma-Aldrich), 150 µM spermine (Sigma-Aldrich)) with 80 µl of zymolyase 20T (15 mg/ml, US Biological) and incubated 35 min at 30˚C to obtain spheroplasts. Pellets were resuspended in 720 µl of $1\times$ TE with 80 µl of µl 10% SDS and incubated for 30 min at 4˚C. DNA was extracted with chloroform:isoamyl alcohol (24:1) and isopropanol. Clean samples were resuspended in 500 µl of $1\times$ TE and treated with 5 µl RNase A (10 mg/ml, Roche) for 90 min at 37˚C. Samples were cleaned again using chloroform:isoamyl alcohol (24:1) and isopropanol. DNA was then digested with 50U SpeI-XhoI (New England Biolabs), and analyzed by Southern blot hybridization into Hybond XL+ membranes (GE Healthcare) with a $^{32P}$-labeled 218-bp *LEU2* probe. The *LEU2* probe was amplified by PCR (Leu Up 2000 and Leu Lo 2000 primers) (*Supplementary file 2*) and purified from agarose gels just before use using a Macherey-Nagel's DNA extraction kit. The signals of the Southern blot membrane were quantified using PhosphorImager Fujifilm FLA-5100 and ImageGauge (Fujifilm) programs. Quantification was performed by calculating the signal corresponding to the DSBs (2.4 and 1.4-kb bands) and SCR+ICR (2.9-kb band) relative to the total DNA in each line from each transformant strain. For every band in the gel, we subtracted the background signal from the same line. The repair ratio was determined by dividing the signal corresponding to SCR+ICR (2.9-kb band) by the sum of the total signal corresponding to DSBs (2.4 and 1.4-kb bands) plus SCR+ICR (2.9-kb band). For the analysis of nicks, DNA samples were additionally subjected to electrophoresis at 4˚C in alkaline conditions (50 mM NaOH, 1 mM EDTA). Nicks were calculated as two times the difference between the media of total breaks (signal in 2.4 and 1.4-kb bands in alkaline gels) and the media of DSBs (signal in 2.4 and 1.4-kb bands in native conditions). Numerical data can be seen in the source data file.

## Genetic analysis of recombination

For the TINV systems, cultures of cells transformed with pTINV-FRT-T, pTINV-FRT-NT, or pTINV-FHO plasmids were grown to mid-log phase in SRaf plasmid-selective media and split in two cultures, one of which was supplemented with doxycycline (5 µg/ml) to repress transcription. Cultures were then split into two again to leave one culture in SRaf (spontaneous recombination frequency) and supplement the other one with 2% galactose to induce FLPm expression during 3 hr (FLPm-induced recombination frequency) or HO during 3 hr (HO-induced recombination frequency). The induction was stopped with 2% glucose and serial dilutions were plated to quantify the number of total or recombinant cells in each case. Leu$^+$ recombinants were selected in SC-leu-ura.

For the DGL-FRT system, cultures of cells transformed either pRS313 or pRS313-GALRNH1 and pCM190-FLP plasmids were grown to mid-log phase in SRaf plasmid-selective media with doxycycline (5 µg/ml) and split into two. One-half was supplemented with 2% glucose or the other half with 2% galactose for 15 min to repress or to induce transcription, respectively. Cultures were then split into two again to leave one culture with doxycycline (spontaneous recombination frequency) and wash the other one three times to allow FLPm expression for 24 hr (FLPm-induced recombination frequency). The reaction was stopped with doxycycline (5 µg/ml) and serial dilutions were plated to quantify the number of total or recombinant cells in each case. Lys$^+$ recombinants were selected in

SGal-ura-lys-his with doxycycline (5 µg/ml). In this case, spontaneous recombination frequencies were obtained from cells transformed with the pCM190.

In all cases, recombination frequencies were calculated as the median value from six independent colonies for each transformant. Numerical data from the mean values obtained for at least three experiments performed with independent transformants can be seen in the source data file.

## Plasmid loss

Cultures of cells transformed with the pCM189-L2FRT-T, pCM189-L2FRT-NT, or pCM189-L2FHO were grown to mid-log phase in SRaf plasmid-selective media and split into two, one of which was supplemented with doxycycline (5 µg/ml) to repress transcription. Cultures were then split into two again to leave one in SRaf (spontaneous plasmid loss) and supplement the other one with 2% galactose to induce FLPm for 24 hr (FLPm-induced plasmid loss) or HO for 1 hr (HO-induced plasmid loss). The reaction was stopped with 2% glucose. Several dilutions were plated in YPAD (to score for total cells) and SC-ura (to score for cells, which have lost the plasmid). Plasmid loss levels were calculated as the median value from six independent colonies for each transformant. Numerical data from the mean values obtained for at least three experiments performed with independent transformants can be seen in the source data files.

## Additional information

### Competing interests

Andrés Aguilera: Reviewing editor, *eLife*. The other authors declare that no competing interests exist.

### Funding

| Funder | Grant reference number | Author |
|---|---|---|
| H2020 European Research Council | ERC2014 AdG669898 TARLOOP | Andrés Aguilera |
| Ministerio de Economía y Competitividad | BFU2016-75058-P | Andrés Aguilera |
| Ministerio de Ciencia, Innovación y Universidades | PDI2019-104270GB-I00 | Andrés Aguilera |
| Junta de Andalucía | P12-BIO-1238 | Andrés Aguilera |
| European Regional Development Fund | | Andrés Aguilera |
| Ministerio de Educación, Cultura y Deporte | PhD FPU fellowship | Pedro Ortega |
| Junta de Andalucía | PhD fellowship | Jose Antonio Mérida-Cerro |
| Junta de Andalucía | P18-FR-566 | Andrés Aguilera |

The funders had no role in study design, data collection and interpretation, or the decision to submit the work for publication.

### Author contributions

Pedro Ortega, Conceptualization, Formal analysis, Validation, Writing - original draft; José Antonio Mérida-Cerro, Conceptualization, Validation; Ana G Rondón, Conceptualization, Supervision; Belén Gómez-González, Conceptualization, Supervision, Investigation, Visualization, Writing - original draft, Writing - review and editing; Andrés Aguilera, Conceptualization, Supervision, Funding acquisition, Project administration, Writing - review and editing

### Author ORCIDs

Pedro Ortega ![ORCID] http://orcid.org/0000-0003-4216-3695
José Antonio Mérida-Cerro ![ORCID] https://orcid.org/0000-0003-4636-5657

Ana G Rondón [ID] http://orcid.org/0000-0002-9481-1255
Belén Gómez-González [ID] https://orcid.org/0000-0003-1655-8407
Andrés Aguilera [ID] https://orcid.org/0000-0003-4782-1714

## Decision letter and Author response

Decision letter https://doi.org/10.7554/eLife.69881.sa1
Author response https://doi.org/10.7554/eLife.69881.sa2

## Additional files

### Supplementary files

- Supplementary file 1. Strains used in this study.
- Supplementary file 2. Primers used in this study.
- Transparent reporting form

### Data availability

All data generated or analysed during this study are included in the manuscript and supporting files. Source data files have been provided for all figures.

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
