## [Decision Letter]

**Acceptance summary:**

The manuscript by Ortega et al. aims to answer a very important question in the field. Specifically, the authors ask whether DNA-RNA hybrids formed at the positions of double-strand breaks (DSBs) promote or interfere recombinational repair of the DSBs. Using plasmid-based systems and a chromosomal system in budding yeast, the authors demonstrate that high levels or unprocessed RNA-DNA hybrids interfere with recombination-mediated DSB repair in plasmids and the chromosome. The authors conclude that RNA-DNA hybrids form accidentally at DSBs in transcribed regions must be removed for productive recombinational repair.

**Decision letter after peer review:**

Thank you for submitting your article "DNA-RNA hybrids at DSBs interfere with repair by homologous recombination" for consideration by *eLife*. Your article has been reviewed by 3 peer reviewers, including Wolf-Dietrich Heyer as the Reviewing Editor and Reviewer #1, and the evaluation has been overseen by Kevin Struhl as the Senior Editor.

The manuscript could be improved by the following text additions and clarifications. No additional experimentation is required but the authors may want to consider adding experimentation to address the concern about the efficiency of break formation in the mutants (#5).

1) Abstract: The abstract should include a sentence about the chromosomal system, as this is critical to generalize the findings with the plasmid-systems.

2) Figure 1: In 1B, which plasmid data are shown? Please expand this part and show spontaneous loss data for all three plasmids.

In 1C, could the author comment on why upon HO break the low transcribed substrate is showing elevated plasmid loss compared to the highly transcribed substrate (in WT background)?

In 1D, could the authors comment on their data in the WT background for replication-dependent DSB on a highly transcribed substrate: RNAseH OE seems to decrease repair efficiency and enhance plasmid loss (2.1 to 14.1), which would argue for a positive function of R-loops in this context. This appears to be different at HO induced breaks, could the authors also comment? Could this indicate that R-loop may be of different nature in these two contexts?

Why the loss is higher for the lower transcription case? How does it support the model? In general, the case of HO breaks looks very different from FRT-T and FRT-NT (Figure 1c), and this includes the frequencies of plasmid loss in all mutants. So, it is unclear how the results shown in these two panels (Figure 1c vs 1d) are consistent with the model.

3) Figure 1: The two FLPm plasmids and the HO plasmid have not been analyzed by DRIP for RNA-DNA hybrids, as have the other plasmid and chromosomal systems. Is there a reason for this? The remaining data are strong and include DRIP data, posing the question whether the effort is worth the gain to add such data here. Maybe this part could be moved to the supplement and a caveat about the lack of DRIP data could be added?

4) Is it possible that hpr1 and rnh1rnh2 mutations affect DSB resection and this could affect the efficiency of repair?

5) How do we know that the efficiency of DSB formation is not affected by various mutations (hpr1 or rnh1rnh2)? Is it possible that accumulation of DSBs reflects not only the efficiency of break formation, but also the kinetics of their repair? How to be sure that the comparison of repair efficiency is conducted under the conditions of equal number of breaks?

6) The addition of the chromosomal system is critical to confirm the plasmid-based data. It would be helpful to the reader to have a few sentences on the caveats of the plasmid system.

7) Figure 2: The normalization of the TINV recombination system should be clarified. I did not fully understand it from the description.

The difference between the highly transcribed and the low transcribed substrates are not really obvious, as the rnh1rhn201 δ strain also displays a slight decrease in Figure 2c. Could the authors add p values compared to the WT for Figure 2B/c?

8) Figure 3: Figure 3a and Figure 3b, left: Why are the levels of R-loops the same in wt with and without transcription (compare the level in a vs b)? Is there another mechanism of RNA/DNA hybrid formation in the absence of promoter? Could it be that PolIII is attracted (in addition to PlII) as it was demonstrated in mammalian cells?

9) Figure 4: The drawing of the repair product for the TINV-HO system is unclear to me. It is the same as Figure 2, where the main inducing lesion is a nick and not a DSB. Shouldn't the product be as drawn in Figure 5 and the product be the same length as the substrate (3.8 kb)? The quantitation and normalization require better description. What are the intermediates, what are/is the product?

---

## [Author Response]

The manuscript could be improved by the following text additions and clarifications. No additional experimentation is required but the authors may want to consider adding experimentation to address the concern about the efficiency of break formation in the mutants (#5).

We thank the editor and reviewers for the positive and constructive comments.

We have now revised the manuscript and extended the figure diagrams and explanations in the text for the repair intermediates and reactions. Given the high fluctuation of the plasmid loss data, we have performed new experiments and a whole re-analysis of all the plasmid loss data.

We have also included a new panel c in Figure 2—figure supplement 2 to strengthen our conclusions.

In addition, we have addressed all specific comments as follows.

1) Abstract: The abstract should include a sentence about the chromosomal system, as this is critical to generalize the findings with the plasmid-systems.

Performed as requested. Thank you.

‘To directly assess whether hybrids formed at DSBs promote or interfere with recombinational repair we have used plasmid and chromosomal-based systems for the analysis of DSB-induced recombination in *Saccharomyces cerevisiae*….// Importantly, we show that high levels or unresolved DNA-RNA hybrids at the breaks interfere with their repair by homologous recombination. This interference is observed for both plasmid and chromosomal recombination and is independent of whether the DSB is generated by endonucleolytic cleavage or by DNA replication.’

2) Figure 1: In 1B, which plasmid data are shown? Please expand this part and show spontaneous loss data for all three plasmids.

This is the result of experiments of spontaneous plasmid loss performed with three transformants with the FRT-T construct and three transformants with the NT construct. However, since there is not break induction, the results can be extrapolated to the HO construct as all three are the same plasmid in which the only variation is the sequence inserted in the leu2 allele (FRT-T FRT-NT or HO). Therefore, we do not expect any difference in the spontaneous source of breaks by changing such short sequences. Since, as stated in the methods section, we preformed these experiments by plating in selective and non-selective media directly after break induction, and not after several rounds of replication in nonselective media, the levels are extremely low (almost undetectable in many cases). Thus, we have now specified the systems used (FRT-T and FRT-NT) and moved this to supplemental (Figure 1—figure supplement 1b) and clarified the text when describing the FRT results (page 6).

In 1C, could the author comment on why upon HO break the low transcribed substrate is showing elevated plasmid loss compared to the highly transcribed substrate (in WT background)?

This could be the result of the increased accessibility or affinity of the HO endonuclease to the HO site at repressed chromatin. Indeed, we previously observed that this is the case and demonstrated that the HO cleavage was 2.5- fold lower with high transcription in a similar plasmid (González-Barrera et al. Genetics 2002, Figure 1c).

We have added this to the text on page 7:

‘Upon transcription activation, however, wild-type cells reduced the frequency of plasmid loss to 8% (Figure 1b). This is in agreement with our previous observation that HO endonuclease activity at the HO site is less efficient at highly transcribed chromatin (González-Barrera et al., 2002).’

In 1D, could the authors comment on their data in the WT background for replication-dependent DSB on a highly transcribed substrate: RNAseH OE seems to decrease repair efficiency and enhance plasmid loss (2.1 to 14.1), which would argue for a positive function of R-loops in this context. This appears to be different at HO induced breaks, could the authors also comment? Could this indicate that R-loop may be of different nature in these two contexts?

We have repeated this experiment with two more transformants and we are now representing the mean of 3 or 4 experiments performed with independent transformants for all the plasmid loss data. As it is now stated in the legends and method section, each experiment was performed with 6 colonies from each transformant, from which we considered the median value, as we do for the recombination tests.

Despite the fact that there is a tendency to an increased plasmid loss when RNase H is overexpressed in the FRT constructs, as mentioned, this is not statistically significant in any of the conditions. This can be caused by the toxic effect of RNase H that has been observed in previous reports (Paulsen et al., 2009; Dominguez-Sanchez et al., 2011). The reason why this is not observed in the HO construct might have to do with the fact that these breaks are replication-independent but also with the fact that RNase H overexpression occurs during 24 hours in the FRT plasmid loss assay and only 1h in the HO. Thus, we modified the text to specify the timing of the experiments on pages 6 and 7 and acknowledged the toxicity of RNAse H overexpression.

Why the loss is higher for the lower transcription case? How does it support the model? In general, the case of HO breaks looks very different from FRT-T and FRT-NT (Figure 1c), and this includes the frequencies of plasmid loss in all mutants. So, it is unclear how the results shown in these two panels (Figure 1c vs 1d) are consistent with the model.

As stated above, we believe that this is due to the higher efficiency of HO cleavage under low transcription conditions that has been previously reported (González-Barrera et al., 2002). It is interesting that this is not the case for the FLP but we believe that this is out of the scope of the manuscript. However, the high plasmid loss observed under low transcription might mask the effect of the mutants. Thus, we decided to repeat the experiment upon longer time of HO induction (3 hours instead of 1h). Under these circumstances, we observed that plasmid loss levels rose up to 70% under low transcription conditions demonstrating that higher levels were possible and detectable in our assay. Moreover, we observed that at high transcription conditions, the rnh1Δ rnh2Δ and hpr1Δ mutants showed more than 50% plasmid loss as compared to the 25% of the wild-type strain. These data are now included in the Results section (page 7) and in Figure 1—figure supplement 1c:

‘Importantly, plasmid loss levels were above 25% in rnh1Δ rnh201Δ and hpr1Δ mutants (Figure 1b). Similar results were obtained after 3h of cleavage-induction, with levels differing from 25% in wt cells to 55% in the mutants (Figure 1–figure supplement 1c).[…] Therefore, whereas rnh1Δ rnh201Δ and hpr1Δ cells led to similar loss levels of cleaved plasmids by either FLPm or HO under low transcription of the plasmidborn leu2 allele, loss frequencies were specifically augmented when leu2 transcription was induced.’

3) Figure 1: The two FLPm plasmids and the HO plasmid have not been analyzed by DRIP for RNA-DNA hybrids, as have the other plasmid and chromosomal systems. Is there a reason for this? The remaining data are strong and include DRIP data, posing the question whether the effort is worth the gain to add such data here. Maybe this part could be moved to the supplement and a caveat about the lack of DRIP data could be added?

We believe that the effort is not worth the gain since the plasmids are similar, except for the additional leu2 repeat that allows us to study repair. We indeed preferred to study the presence of hybrids in the same systems in which we study directly repair and not only plasmid loss. Admitting that it is a possibility, we prefer not moving figure 1 to supplemental since it is the start point for the two types of breaks covered in the study (replication-born and induced by endonucleolitic cleavage). We have now started the second section of results as follows:

“To directly study DSB repair upon DNA-RNA hybrid accumulation, we took advantage of the TINV recombination system.”

and the third section of results as follows:

“To assay whether DNA-RNA hybrids accumulate upon break induction in our repair systems”

4) Is it possible that hpr1 and rnh1rnh2 mutations affect DSB resection and this could affect the efficiency of repair?

Certainly, it is possible that the DNA-RNA hybrids accumulated affect resection, since, as discussed on page 12, persistent hybrids caused by depletion of RNase H, helicases or the human exosome have been shown to affect RPA binding and/or DNA end resection in other yeast or human cell studies (Daley et al., 2020; Domingo-Prim et al., 2019; Li et al., 2016; Ohle et al., 2016; Rawal et al., 2020; Sessa et al., 2021; Yu et al., 2020). We have now added the following sentences to the Discussion section to state this possibility clearly. Thank you.

“It is therefore possible that DNA-RNA hybrid accumulation in hpr1Δ and rnh1Δ rnh201Δ affects repair by negatively interfering with DSB resection. This could be particularly important when the hybrid covers the 5’ end that needs to be resected, although we have observed the formation of break-induced DNA-RNA hybrids at both sides of the break (Figure 3, Figure 3—figure supplement 1, Figure 5, Figure 5—figure supplement 1, Figure 6 and Figure 6—figure supplement 1). The hybrid formed in the 3’ end might initially favor resection of the complementary strand as we previously proposed (Aguilera and Gómez-González, 2017), but we envision that it would need to be removed later to allow the loading of RPA and subsequently Rad51.”

5) How do we know that the efficiency of DSB formation is not affected by various mutations (hpr1 or rnh1rnh2)? Is it possible that accumulation of DSBs reflects not only the efficiency of break formation, but also the kinetics of their repair? How to be sure that the comparison of repair efficiency is conducted under the conditions of equal number of breaks?

We cannot establish the efficiency of DSB formation itself but the levels of molecules with nicks or DSBs, which are certainly the result of break formation and repair at each time point. This is indeed the reason why DSBs accumulate at later time points in the hpr1 and rnh1rnh2 mutants, in which we believe that defective repair is occurring. To study this reaction, we estimate the repair efficiency as the ratio between repaired molecules and total molecules that are either cleaved or repaired. We have tried to explain this better on page 7 now as follows:

“The levels of DSBs detected were higher in rnh1Δ rnh201Δ and hpr1Δ mutants (Figure 2—figure supplement 1c-d). This could be a consequence of either a major efficiency of breakage or a lower efficiency of DSB repair, which in these FRT-based constructs occurs preferentially by SCR (Ortega et al., 2019).”

6) The addition of the chromosomal system is critical to confirm the plasmid-based data. It would be helpful to the reader to have a few sentences on the caveats of the plasmid system.

We have added this information to page 11. Thank you.

“Although plasmid systems have been recurrently validated as models to study DNA repair and recombination, we wanted to confirm our conclusions in chromosomal DSBs to make sure that any putative local difference in chromatin or topology, even though unlikely, did not affect results.”

7) Figure 2: The normalization of the TINV recombination system should be clarified. I did not fully understand it from the description.

As stated above, we have explained this better on page 7 now and also in the figure legend. Thank you.

The difference between the highly transcribed and the low transcribed substrates are not really obvious, as the rnh1rhn201 δ strain also displays a slight decrease in Figure 2c. Could the authors add p values compared to the WT for Figure 2B/c?

We added the information about p-value of two-way Anova tests in the figure 2 legend (p > 0.1 in all cases) and stated in the text that the differences were not significant. On page 7, it now reads:

“Under high transcription of the FRT site, a subtle but not significant defect in repair was detected in rnh1Δ rnh201Δ cells in both FRT-T and FRT-NT constructs and in hpr1Δ in the FRT-T construct (Figure 2b). Interestingly, such a tendency was not observed under low transcription (Figure 2c) suggesting that although subtle, there could be a repair defect that was transcriptiondependent.”

8) Figure 3: Figure 3a and Figure 3b, left: Why are the levels of R-loops the same in wt with and without transcription (compare the level in a vs b)?

These are backup levels of hybrids that were not induced by the break. There is no evidence that hybrids form spontaneously at this short leu2 gene, unless Hpr1 is removed, as we see here. In Figure 3b, the signals are not even reduced by RNase H treatment, likely reflecting that they are not hybrids but background signal.

Is there another mechanism of RNA/DNA hybrid formation in the absence of promoter? Could it be that PolIII is attracted (in addition to PlII) as it was demonstrated in mammalian cells?

This is an interesting possibility, but we have not observed any break-induced hybrids in the absence of transcription in either of the FRT systems (plasmid in Figure 3b or chromosomal system in Figure 6e). In contrast, some breakinduced hybrids are still present upon transcription inhibition in the HO system, although at lower levels as compared to high transcription conditions (Figure 5e). Thus, we favor the idea that they reflect the leaky transcription from the tet promoter upon transcriptional repression (Figure 1—figure supplement 1a) when levels of breaks are high as those obtained with the HO endonuclease. To be more accurate in the discussion, we have now changed the text as follows:

“However, we show that this phenomenon is completely dependent on the induction of transcription of the construct driven by the TET promoter in the case of the FLP-induced breaks (Figures 3 and 6) and partially reduced in the case of HO-induced breaks (Figure 5). Therefore, we conclude that DNA breaks lead to DNA-RNA hybrids at both sides of the break due to pre-existing ongoing transcription rather than de novo RNA synthesis previously proposed (Ohle et al., 2016; Rawal et al., 2020).”

9) Figure 4: The drawing of the repair product for the TINV-HO system is unclear to me. It is the same as Figure 2, where the main inducing lesion is a nick and not a DSB. Shouldn't the product be as drawn in Figure 5 and the product be the same length as the substrate (3.8 kb)? The quantitation and normalization require better description. What are the intermediates, what are/is the product?

We have re-drawn and re-described the systems in figures 2 and 4. Thank you. On page 8, when describing the TINV-FRT systems, we have added:

“To determine the frequency of SCR we quantified the events involving an exchange between unequal repeats in the two sister chromatids (unequal sister chromatid exchange, uSCE), which leads to a dicentric dimer intermediate that can be visualized a by the 4.7 and 2.9-kb bands resulting from XhoI and SpeI digestion (Figure 2a) (González-Barrera et al., 2003). Given the proximity of the 4.7-kb band to the strong 3.8-kb band arising from the digestion of the more abundant intact plasmid, we relied on the 2.9-kb band to quantify SCR as previously described (Ortega et al., 2019). Other recombination reactions are also possible but known to occur at a minor and irrelevant frequency (Cortés-Ledesma et al., 2007). Thus, to estimate repair at each time-point, we calculated the ratio between the SCR-derived molecules (2.9-kb fragment, Figure 2a) and the sum of repaired and cleaved molecules (2.9-kb, 2.4 and 1.4-kb fragments, Figure 2a) (see Methods).”

And on page 11, when describing the TINV-FO system, we have modified the text as follows:

“As in the previous systems, the 2.4 and 1.4-kb bands corresponded to DSBs and the 2.9-kb band to HR repair intermediates, which in this case would mostly result from BIR initiated from one of the DSB ends invading the truncated repeat located in the same chromatid, an intrachromatid recombination (ICR) reaction (Figure 4a). This is so because an intact sister chromatid for SCR would only be present in G2 when only one of the chromatids was cleaved by chance, and we determined the HR repair ratios (Figure 4 and Figure 4—figure supplement 1).”